# Evolving RL: Discovering New Activation Functions using LLMs

## Abstract

Deep Reinforcement Learning (DRL) has traditionally inherited activation functions from supervised learning, despite fundamental differences in learning dynamics and objectives. We present EvolveAct, a novel framework that leverages large language models and evolutionary search to automatically discover optimal activation functions for specific RL tasks. Our method combines genetic programming with code Large Language Models (LLMs) to explore a rich space of mathematical functions, optimizing for stability and performance in DRL training. Experimental results across multiple environments show that the discovered activation functions consistently outperform standard choices such as ReLU and TanH, improving final performance on the Minatar suite by 37.25% and 28.3% on the Brax suite on average. By jointly optimizing over multiple diverse environments, we discover activation functions that demonstrate strong generalization capabilities across different RL domains. This research provides a foundation for automating fundamental architectural choices in deep reinforcement learning systems.

## 1 Introduction

Deep Reinforcement Learning has seen tremendous success in recent times, ranging from superhuman performance in Chess & Go Silver et al. (2017), to powering robotics, and now is even the driving force behind emergent reasoning in LLMs Liu et al. (2024). Despite these advances, DRL has largely inherited its fundamental building blocks from supervised learning, including crucial components like activation functions. While this transfer of knowledge has provided a strong foundation, these inherited design choices can lead to instability and brittle performance in RL settings, where small changes in training setup can cause significant degradation Henderson et al. (2018); Chan et al. (2019). The unique challenges of RL, such as non-stationarity and bootstrap-based training, suggest we should question these inherited design choices.

In this work, our aim is to further explore the choice of activation functions in DRL. Activations are at the heart of neural networks, transforming simple linear models into powerful universal function approximators. Therefore, the choice of activation function greatly influences learning dynamics-shaping how gradients flow through the network. For instance, Bhatt et al. (2019) demonstrates that bounded activations like tanh eliminate the need for target networks, dramatically improving sample efficiency. Similarly, alternative activations like concatenated ReLU have shown significant gains in addressing issues like dormancy Sokar et al. (2023) and saturation Kooi et al. (2024) in non-stationary tasks like RL.

These findings hint at a vast unexplored space of activation functions, each potentially offering unique benefits for different RL scenarios. However, discovering these functions remains a challenge. Unlike numerical hyperparameters that can be optimized through standard methods, activation functions represent complex mathematical mappings whose design space is infinitely rich. This raises an intriguing question: Could there exist activation functions, yet undiscovered, that are particularly well suited for specific RL tasks?

Our key contributions are as follows:

- **Expanding the search for activation functions:** We show that the space of high-performing activation functions remains largely unexplored. Simply changing the activation function can yield significant performance improvements over standard choices.

- **Introducing EvolveAct:** We present a novel method that combines evolutionary search with large language models to discover activation functions optimized for different RL tasks.

## 2 Method: Evolutionary Discovery of Activation Functions

Our approach combines evolutionary search with LLMs to discover novel activation functions for reinforcement learning. Defining an evolutionary search process typically involves defining three key components: population initialization, fitness evaluation, and an LLM-guided evolutionary process.

### 2.1 Population Initialization

We initialize our population with foundational activation functions that are popularly used across Deep Learning: ReLU, Tanh, Sigmoid, and Leaky-ReLU. While domain-specific activations like Swish Ramachandran et al. (2017), PELU Godfrey (2019), C-ReLU Shang et al. (2016), etc. exist, we exclude these from the initial population as they are typically compositions of simpler functions and hence can be discovered from crossovers of the base population.

### 2.2 Fitness Evaluation

To evaluate the fitness of an activation function, we integrate it into both actor and critic networks and conduct RL training across multiple random seeds. The fitness of an activation is defined as the cumulative reward obtained throughout training, averaged across seeds. While alternative metrics like final performance could be used (as in Goldie et al. (2024)), we find that cumulative reward better captures both learning speed and training stability. Our empirical observations show that activation functions that improve early training consistently maintain or improve final performance, making this metric particularly suitable.

### 2.3 LLM-Guided Evolution

The core innovation of our approach lies in using LLMs to perform intelligent crossovers between activation functions. Traditional genetic programming requires careful design of crossover operators to maintain syntactic validity while ensuring sufficient diversity in the search space Chen et al. (2024); Nader & Azar (2021). We circumvent this design challenge by leveraging LLMs' code generation capabilities, providing pairs of parent activation functions along with their fitness scores, prompting the model to generate novel combinations (detailed prompt in Appendix A). The complete evolutionary process proceeds as follows:

1. Initialize the population with standard activation functions and evaluate their fitness

2. For each round of evolution:

    (a) Sample M pairs of activation functions from the current population

    (b) Generate N new activation functions per pair using LLM-guided crossover

    (c) Evaluate the fitness of all M×N new functions.

    (d) Select the top K performing functions to form the next generation.

3. Repeat step 2 for a fixed number of rounds.

This process combines the exploration capabilities of evolutionary algorithms with the structured knowledge embedded in LLMs to efficiently search the space of activation functions. For a detailed pseudocode of the algorithm, refer to Appendix F.

## 3 MULTI ENVIRONMENT DISCOVERY

The optimal activation function for deep reinforcement learning is both algorithm and environment specific, due to varied learning dynamics and environment properties. In our previous section we detailed an algorithm to discover environment-specific activation functions, a key challenge is finding activation functions that generalize well across multiple environments. This section presents our approach to discovering activation functions that perform robustly across diverse reinforcement learning tasks, even if they may not be optimal for any single environment.

### 3.1 FITNESS FUNCTION DESIGN

To evaluate activation functions across multiple environments, we need a carefully designed fitness function that accounts for the varying scales of returns across different environments. A naive approach of summing individual environment fitness scores would be problematic, as environments with larger return ranges would dominate the optimization process. Instead, we propose a normalized fitness score that gives equal weight to relative improvements across all environments. Our multi-environment fitness function is defined as:

$$\sum_{env \in \mathcal{E}} \frac{f_{new}(env) - \max_{base \in \mathcal{B}} f_{base}(env)}{|\max_{base \in \mathcal{B}} f_{base}(env)|} \tag{1}$$

Here, $\mathcal{E}$ represents the set of target environments and $\mathcal{B}$ represents the set of baseline activation functions in the initial population. The terms $f_{new}(env)$ and $f_{base}(env)$ denote the performance of the candidate activation function and baseline activation function on environment $env$, respectively.

This formulation normalizes the improvement of each new activation function relative to the best performing baseline activation for that specific environment.

### 3.2 COMPOSITE REWARD

While our normalized fitness function helps prevent dominance by high-return environments, we observed that it could still lead to optimization favoring environments where improvements are easier to achieve, potentially neglecting more challenging environments where gains are harder to obtain. To address this limitation, we introduce a composite reward structure that explicitly considers both the breadth and magnitude of improvements across environments. We define the fitness of an activation function as an ordered pair, where the first component measures the number of environments showing improvement, and the second component is the normalized score from Equation 1. More formally, let $F_i$ denote the normalized improvement for environment $i$, defined as:

$$F_i = \frac{f_{new}(env_i) - \max_{base \in \mathcal{B}} f_{base}(env_i)}{|\max_{base \in \mathcal{B}} f_{base}(env_i)|} \tag{2}$$

The composite fitness function is then defined as:

$$\left( \sum_{i \in \mathcal{E}} \not\Vdash (F_i > 0), \sum_{i \in \mathcal{E}} F_i \right) \tag{3}$$

where $\not\Vdash_{F_i > 0}$ is an indicator function that returns 1 if $F_i$ is positive and 0 otherwise. We compare activation functions using lexicographic ordering on these pairs. This approach ensures that our search prioritizes activation functions that provide consistent improvements across many environments over those that might achieve larger gains in just a few environments.

## 4 EXPERIMENTAL DETAILS AND RESULTS

We evaluate our approach on two distinct suites of environments: Brax Freeman et al. (2021); Todorov et al. (2012), a widely used continuous control benchmark, and Minatar Young & Tian (2019); Lange (2022). Both implementations leverage the computational efficiency of end-to-end JAX pipelines in reinforcement learning Lu et al. (2022).

Consistent with established practices in meta-learning and algorithmic discovery research Goldie et al. (2024); Bingham et al. (2020); Chen et al. (2023), we maintain fixed evolutionary strategy

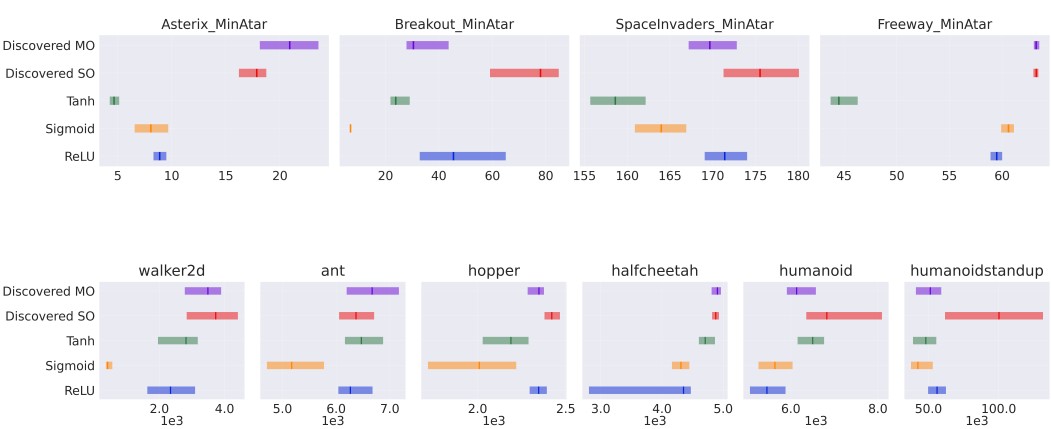

Figure 1: IQM of mean final return for different activations across different environment suites (a) Minatar and (b) Brax with 95% confidence intervals.

parameters due to computational constraints. All experiments utilize the PPO algorithm Schulman et al. (2017) with default hyperparameters from the PureJaxRL library Lu et al. (2022), and reported performance metrics represent averages across 16 independent seeds. Detailed hyperparameters for both evolutionary search and RL training are provided in Appendix D.

Our experimental evaluation consists of two primary investigations. First, we demonstrate the effectiveness of single-objective optimization on individual environments. Second, we evaluate our multi-environment optimization approach (Section 3.2) by training collectively on all Brax environments and, separately, all Minatar environments. Results from both approaches are presented in Figures 4a and 4b, with the specific discovered activation functions detailed in Appendix E.

While our optimization targets total return, we present Interquartile Mean (IQM) plots of final returns for clearer visualization of performance improvements. In the Brax suite, our approach achieves average improvements over ReLU of **27.58%** and **10.11%** for single-objective and multi-objective optimization, respectively. The Minatar environments demonstrate even more substantial gains, with improvements of **37.21%** and **28.3%**.

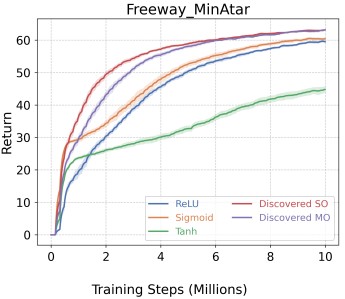

Figure 2: Training curves for PPO using various activations. Our discovered activations demonstrate better sample complexity and stability.

It is worth noting that while some environments do not exhibit substantial improvements in final performance, our approach demonstrates significant advantages in terms of sample efficiency and training stability. This improvement is a natural consequence of optimizing for cumulative return across the entire training trajectory, rather than solely emphasizing final reward values. Figure 2 illustrates this phenomenon, where despite similar final performance across methods, our approach achieves notably faster convergence and higher cumulative return throughout training. Complete training curves for all environments are provided in Appendix C.

## 5 CONCLUSION AND FUTURE WORK

In this work, we demonstrated that automatically discovering RL-optimized activation functions can significantly improve performance across diverse environments. With increased computational resources, our results suggest potential for discovering general-purpose activation functions for RL. This work opens promising directions for future research, from quality diversity algorithmsNasir et al. (2024); Pugh et al. (2016) to advanced evolutionary strategies and leveraging LLM capabilities like in-context learning Lu et al. (2024). We believe automated discovery of neural network components will become increasingly vital for optimizing RL systems.

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

# A    PROMPT STRATEGY

The exact prompt that we use to generate a new activation is the following:

---

**Context:**
You are assisting in discovering novel activation functions for reinforcement learning. You will be given examples of previous activation functions and their performance scores and should propose new functions that could perform better.

---

**Previous Functions and Scores: Function 1:**

```
[FUNCTION_CODE_1]
```

Score: [SCORE_1]

**Function 2:**

```
[FUNCTION_CODE_2]
```

Score: [SCORE_2]

---

**Requirements:**

- First provide your reasoning for the proposed function design.
- Then provide the implementation in JAX with necessary imports.
- The function should:
    - Take a single input parameter $x$.
    - Return a corresponding output.
    - Be differentiable.
    - Be implemented using JAX operations.
    - Ensure the function is named as `jax_activation`.
- Keep the code clear and extractable.

---

**Desired Response Format:**
**Reasoning:** [Explain your approach and why it might perform better, based on the previous examples' scores and general principles of activation functions]

**Implementation:**

```
import jax.numpy as jnp
from jax import nn  # Include any other necessary imports

def jax_activation(x):
    return [your implementation]
```

---

## B    RELATED WORK

### B.1    ACTIVATIONS IN RL

The ReLU (Rectified Linear Unit) Maas et al. (2013) activation function dominates supervised learning, but it suffers from a significant drawback in non-stationary learning tasks like RL: the "dead neuron" problem Sokar et al. (2023) where neurons can become permanently inactive. While tanh (hyperbolic tangent) has been used as an alternative, it too has limitations, particularly the saturation effect that can slow down learning Kooi et al. (2024). These challenges have spurred research into more effective activation functions to enhance neural plasticity and improve reinforcement learning performance.

Recent advances in this area include several promising approaches. Kooi et al. (2024) proposed a modification to the tanh activation that shows significant improvements. Among others, three notable innovations have also emerged: PELU (Parametric Exponential Linear Unit) Godfrey (2019), a learnable variant of ELU Clevert (2015) that allows the network to adapt its activation characteristics during training; CReLU (Concatenated ReLU) Shang et al. (2016), a technique that preserves information by concatenating both positive and negative activations; and adaptive rational activations, an approach that uses ratios of polynomials to create flexible, learnable activation functions Delfosse et al. (2021).

#### B.1.1    ACTIVATION DISCOVERY

While the most widely used activation functions remain handcrafted, prior work has successfully discovered novel activations for supervised learning. Notably, Swish activation was discovered through reinforcement learning-based search and has found applications in computer vision Ramachandran et al. (2017). Alternatively evolutionary approaches Bingham & Miikkulainen (2022); Nader & Azar (2021) have also demonstrated the potential of automatically discovering activation functions for supervised learning tasks.

### B.2    EVOLUTIONARY DISCOVERY THROUGH LLMS

The integration of Large Language Models (LLMs) with evolutionary algorithms has sparked a renaissance in evolutionary computation Wu et al. (2024). By redefining crossover operations in more intuitive ways, LLMs have made evolutionary methods more accessible and effective across diverse applications. A landmark example is FunSearch Romera-Paredes et al. (2024), which successfully leverages this combination to discover novel heuristics for computationally challenging NP-Hard problems. This approach has also shown promise in neural architecture search, where researchers have demonstrated that evolutionary strategies enhanced by LLMs can effectively optimize neural network architectures Chen et al. (2023); Nasir et al. (2024). The success of these methods suggests that LLMs can serve as powerful tools for guiding evolutionary search processes in complex solution spaces.

## C    TRAINING CURVES

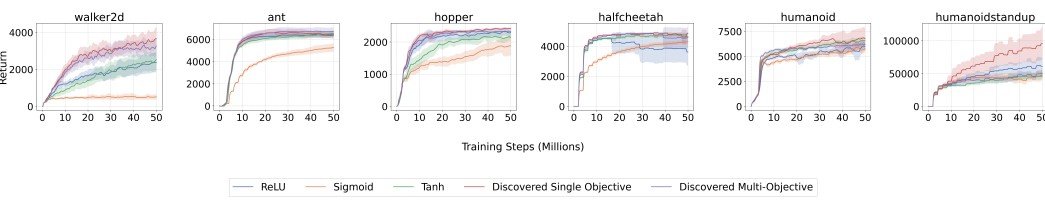

Figure 3: Training curves for PPO using various activation functions in the Brax Suite. Results are averaged over 16 random seeds, with 95% confidence intervals shown.

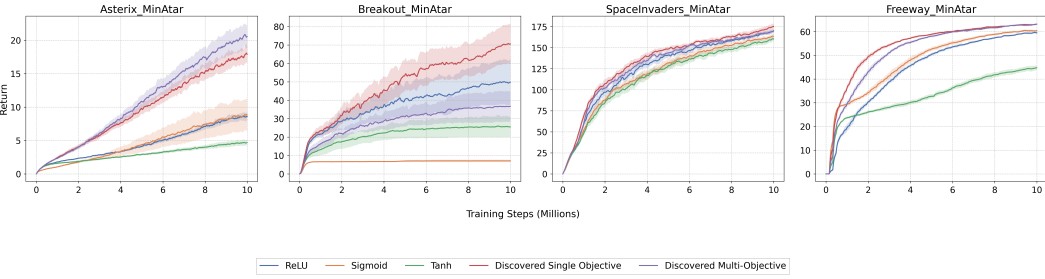

Figure 4: Training curves for PPO using various activation functions in the Minatar Suite. Results are averaged over 16 random seeds, with 95% confidence intervals shown.

## D HYPERPARAMETERS USED

### D.1 HYPERPARAMETERS FOR RL TRAINING ON MINATAR

| | |
|---|---|
| **Learning Rate** | 0.005 |
| **Number of Environments** | 64 |
| **Number of Steps** | 128 |
| **Total Timesteps** | 10,000,000 |
| **Update Epochs** | 4 |
| **Number of Minibatches** | 8 |
| **Gamma** | 0.99 |
| **GAE Lambda** | 0.95 |
| **Clip Epsilon** | 0.2 |
| **Entropy Coefficient** | 0.01 |
| **Value Function Coefficient** | 0.5 |
| **Max Gradient Norm** | 0.5 |
| **Anneal Learning Rate** | True |
| **Number of Seeds** | 16 |

### D.2 HYPERPARAMETERS FOR RL TRAINING ON BRAX

| | |
|---|---|
| **Learning Rate** | 3.0e-4 |
| **Number of Environments** | 2048 |
| **Number of Steps** | 10 |
| **Total Timesteps** | 50,000,000 |
| **Update Epochs** | 4 |
| **Number of Minibatches** | 32 |
| **Gamma** | 0.99 |
| **GAE Lambda** | 0.95 |
| **Clip Epsilon** | 0.2 |
| **Entropy Coefficient** | 0.0 |
| **Value Function Coefficient** | 0.5 |
| **Max Gradient Norm** | 0.5 |
| **Anneal Learning Rate** | False |
| **Normalize Environment** | True |
| **Number of Seeds** | 16 |

## D.3 Hyperparameters for Evolution

| | |
|---|---|
| **LLM Model** | Gemini-1.5-flash |
| **Number of Phases** | 10 |
| **Number of Prompts** | 15 |
| **Number to Keep** | 15 |
| **Number of Samples** | 2 |

## E Discovered Activations

| Suite | Optimized Activation Function |
|---|---|
| Brax | $\sigma(x)\tanh(x) + (1 - \sigma(x))\max(0, x)\sigma(x)$ |
| MinAtar | $\mathrm{softplus}(x) + 0.1\tanh(2x)$ |

Table 1: Activation functions optimized across entire suites using multi-objective optimization. $\sigma(\cdot)$ denotes the sigmoid function.

| Environment | Activation Function |
|---|---|
| Ant | $\begin{cases} x & \text{if } x \geq 0 \\ 0.1 \cdot \mathrm{softplus}(x) \cdot x & \text{otherwise} \end{cases}$ |
| Asterix | $0.1 \cdot \max(0, x) + 0.4 \cdot \sigma(x - 1.2) \cdot x$ |
| Breakout | $\begin{cases} x & \text{if } x \geq 0 \\ x \cdot \tanh(x) & \text{otherwise} \end{cases}$ |
| Freeway | $\begin{cases} \mathrm{softplus}(1.0x) & \text{if } x \geq 0 \\ \mathrm{softplus}(0.1(-x)) & \text{otherwise} \end{cases}$ |
| HalfCheetah | $\begin{cases} 1 + 0.01(x - 1) & \text{if } x > 1 \\ x + 0.001x & \text{if } x \geq 0 \\ 0.1(x\sigma(x) + \sin(2x)) + 0.01x & \text{otherwise} \end{cases}$ |
| Hopper | $x \cdot \sigma(x/\alpha)$ where $\alpha = 0.1$ |
| Humanoid | $\mathrm{softplus}(x) - \ln(2) + 0.1\exp(-x^2/8)$ |
| Humanoid Standup | $\mathrm{softplus}(x) + 0.1\tanh(2x) + 0.01x^3 + 0.001e^{-|x|}$ |
| Space Invaders | $\alpha(x \cdot \sigma(x)) + (1 - \alpha)\mathrm{softplus}(x)$ |
| Walker | $\frac{x\mathrm{softplus}(x)\sigma(x)(x + 0.1x^3)}{1 + 0.1|x|}e^{-x^2/2} + 0.0001x^3\sigma(x)$ |

Table 2: Environment-specific activation functions. $\sigma(\cdot)$ denotes the sigmoid function, and $\alpha$ is a hyperparameter where not explicitly specified.

# F  PSEUDOCODE OF ALGORITHM

---

**Algorithm 1** EvolvAct: LLM-guided Evolution of Activation Functions

---

**Require:** Initial population size K, number of pairs M, variants per pair N, number of rounds R
**Require:** Large Language Model LLM, Fitness evaluation function F

1: $P \leftarrow \{\text{ReLU}, \text{Tanh}, \text{Sigmoid}, \text{LeakyReLU}\}$ {Initialize population}
2: $scores \leftarrow \{f \mapsto F(f) \mid f \in P\}$ {Evaluate initial population}
3: **for** round = 1 to R **do**
4:     $pairs \leftarrow \text{SamplePairs}(P, scores, M)$ {Sample M pairs}
5:     $new\_functions \leftarrow \{\}$
6:     **for** $(f_1, f_2)$ in pairs **do**
7:        **for** i = 1 to N **do**
8:           $f_{new} \leftarrow \text{LLM.generate\_crossover}(f_1, f_2, scores[f_1], scores[f_2])$
9:           $new\_functions \leftarrow new\_functions \cup \{f_{new}\}$
10:        **end for**
11:     **end for**
12:     $new\_scores \leftarrow \{f \mapsto F(f) \mid f \in new\_functions\}$
13:     $P \leftarrow \text{TopK}(P \cup new\_functions, scores \cup new\_scores, K)$ {Keep top K}
14:     $scores \leftarrow \{f \mapsto scores[f] \mid f \in P\}$
15: **end for**
16:
17: **return** $\arg\max_{f \in P} scores[f]$ {Return best activation function}

---

