# OpenReview forum: "Evolving RL: Discovering New Activation Functions using LLMs"
_ICLR.cc/2025/Workshop/AgenticAI — ICLR 2025 Workshop AgenticAI Poster_

### Official Review · Reviewer_sJQz · 2025-03-03
**Review of Evolving RL: Discovering New Activation Functions using LLMs**

**Rating:** 6
**Confidence:** 4

**Review:**

Pros:
1. This paper is well-written. The motivation is clear, and the paper is very easy to follow.
2. The experiments are very promising.
3. The activation functions this paper finds are relatively simple, not too complicated. They seem to be practical in real implementation.

Cons:
1. The experiments are only conducted on PPO with fixed hyperparameters but have not been applied to other RL algorithms. It would be great to conduct more experiments in broader settings.
2. The evolutionary process likely requires significant computational resources to evaluate multiple activation functions across environments and random seeds, though specific requirements aren't fully detailed.

---

### Official Review · Reviewer_CSv6 · 2025-03-05
**Good Paper**

**Rating:** 7
**Confidence:** 4

**Review:**

This paper introduces EvolveAct, a framework that automatically discovers new activation functions for Deep Reinforcement Learning (DRL) using large language models (LLMs) and evolutionary search. Traditional DRL models use standard activation functions like ReLU and Tanh, which were originally designed for supervised learning, despite fundamental differences in learning dynamics. EvolveAct initializes with basic activation functions and evolves new ones through a combination of genetic programming and LLM-guided crossover operations. The paper suggests that custom activation functions tailored to specific RL tasks can significantly enhance performance, opening the door for automated architectural optimizations in deep learning.

Strengths of the Paper:
1. Automates the discovery of activation functions, reducing the reliance on manual design.
2. Leverages LLMs for intelligent function generation, improving diversity in search.
3. Demonstrates significant performance gains over traditional activation functions in RL.

Weaknesses of the Paper:
1. Dependent on LLM-generated functions, which may introduce bias or suboptimal solutions.
2. Computationally expensive, multiple RL training runs are required to evaluate each activation function.

---

### Decision · Program_Chairs · 2025-03-05

Accept (Poster)